# Landslide Displacement Prediction Based on Variational Mode Decomposition and GA–Elman Model

**Wei Guo** [1,2], **Qingjia Meng** [1,2], **Xi Wang** [1,2], **Zhitao Zhang** [3], **Kai Yang** [1,2,*] and **Chenhui Wang** [1,2,*]

1   Technology Innovation Center for Geological Environment Monitoring, Ministry of Natural Resources, Baoding 071051, China
2   Center for Hydrogeology and Environmental Geology Survey, China Geological Survey, Baoding 071051, China
3   School of Business Administration, Northwest A&F University, Yangling, Xianyang 712100, China
*   Correspondence: yangkai@mail.cgs.gov.cn (K.Y.); wangchenhui@mail.cgs.gov.cn (C.W.)

**Abstract:** Landslide displacement prediction is an important part of monitoring and early warning systems. Effective displacement prediction is instrumental in reducing the risk of landslide disasters. This paper proposes a displacement prediction model based on variational mode decomposition and a genetic algorithm optimization of the Elman neural network (VMD–GA–Elman). First, using VMD, the landslide displacement sequence is decomposed into the three subsequences of the trend term, the periodic term, and the random term. Then, appropriate influencing factors are selected for each of the three subsequences to construct input datasets; the rationality of the selection of the influencing factors is evaluated using the gray correlation analysis method. The GA–Elman model is used to forecast the trend item, periodic item and random item. Finally, the total displacement is obtained by superimposing the three subsequences to verify the performance of the model. A case study of the Shuizhuyuan landslide (China) is presented for the validation of the developed model. The results show that the model in this paper is in good agreement with the actual situation and has good prediction accuracy; it can, therefore, provide a basis for early warning systems for landslide displacement and deformation.

**Keywords:** landslide displacement prediction; variational mode decomposition; genetic algorithm; Elman neural network; gray relational analysis

## 1. Introduction

Landslides cause significant casualties and property damage each year around the world [1–3]. With the rapid development of the global economy and society, and the concomitant intensification of climate extremes and human engineering activities, landslide disasters increase in frequency day by day. Notably, many large hydropower plants and reservoirs have been built around the world to reduce air pollution and carbon emissions from coal power stations; this often leads to landslides around the reservoirs [4,5]. Such landslides are extremely harmful, not only affecting the safety of reservoirs and hydropower stations, but also impacting the surrounding residents, ships in the rivers, and the surrounding roads [6,7]. Therefore, large reservoir landslides are a pressing problem for geological disaster prevention and reduction. It is of great significance to make reasonable and accurate predictions of land displacement in large reservoir areas.

The process whereby landslides develop in large reservoir areas is a result of the interaction of many factors, which exhibit complex and nonlinear characteristics [8]. These factors can be divided into two categories: internal and external [9–11]. The internal factors include changes in landslide stress and geometry, while the external factors include local precipitation, changes in reservoir water levels, and snow melting. Due to the complex processes of geological change and the conditions of landslide formation, it remains challenging to accurately predict landslide displacement; as such, much geological

research has been devoted to this issue. Since the Japanese scholar Saito Masahiro proposed the empirical prediction formula for landslide displacement in the 1960s, many scholars have devoted their efforts to the field of landslide displacement prediction, achieving some significant results [12–15]. Intrieri et al. systematically summarized the different types of landslide displacement prediction methods, classified these methods, and discussed their respective differences and characteristics [16]. At present, landslide displacement prediction models can be roughly divided into the following four types:

(1) The empirical prediction model. This kind of method uses strict derivation methods, such as mathematics and physics, to analyze a large number of landslide monitoring data and experimental data; this is combined with the transformation of relevant formulas to predict the occurrence of landslide displacement. Focusing on the intrinsic causes of landslide displacement, various parameters of landslides are expressed numerically and according to relevant mathematical formulas. Representative models include the Saito model (1965), the HOCK method (1977), and the Crosta and Agliardi model (2012). However, the scope of application of the model is greatly limited by the lack of understanding of the nature of landslides; moreover, the prediction accuracy of the model is not high.

(2) The statistical prediction model. This method uses the theoretical knowledge derived from modern mathematics to design a landslide prediction model. In contrast to the empirical prediction period, which focuses on the mathematical expression of the landslide's own mechanisms [17], this method includes an investigation and statistical analysis of the geological environment surrounding the landslide, as well as the external factors. At the same time, the prediction accuracy and application scope of the model are also significantly improved at this stage. The rapid development of the statistical prediction model is attributed to the emergence and widespread application of modern mathematical theories, such as mathematical statistics, gray system theory, and probability theory. In recent years, many new theories and methods have been formed. For example, Xu et al. (2011) introduced the GM (1,1) model of gray system theory into the field of landslide displacement [18]. In addition, there are gray vector angle models, models based on landslide slope changes [19,20], etc. Most of these models are linear models, which show better results in predicting the displacement of landslides affected by a single factor, but have poor predictive effects for landslides with complex causes and many influencing factors.

(3) The nonlinear prediction model. With the development and widespread application of system science and nonlinear science, scholars have realized that landslides are an open and complex system. To predict a landslide, qualitative discrimination and quantitative prediction must be combined to study the basic problems that lead to a landslide. Qualitative discrimination refers to the combination of precursor features, such as those exhibited prior to the evolution of the landslide, and the surrounding geological environment [21] Quantitative prediction refers to the quantitative analysis of the observed landslide displacement information data. During this period, BP and Elman neural network models were widely used [22–26]. The extreme learning machine model and the decision tree model have also been gradually introduced and applied to the field of landslide prediction [5,27,28].

(4) The comprehensive prediction model. When using a single nonlinear model to predict landslides, the application range and prediction accuracy of the model are sometimes limited [9]. In recent years, the comprehensive use of multiple models has become a new trend in the development of landslide prediction models. For instance, Miao et al. proposed a landslide displacement prediction model based on GA–SVR [29], while Zhang et al. studied the WCA–ELM model, which is applicable to step-type landslides [30]. Methods for the decomposition of displacement data include empirical mode decomposition [31,32], ensemble empirical mode decomposition [33–35], and variational mode decomposition [36,37]. Although these methods can completely decompose the data and effectively improve prediction accuracy, the physical meaning

of each component cannot be clarified due to the large number of components acquired (generally more than five); as such, they cannot effectively reflect the relationship between each displacement component and the influencing factors [38]. The Elman neural network has good dynamic characteristics and global stability, and it has been widely used to analyze and process nonlinear and dynamic complex data. Chen et al. (2017) verified the feasibility of using the Elman neural network model in landslide monitoring and prediction [39]. Taking into account the nonlinear characteristics of landslide displacement monitoring data, they proposed an improved recurrent neural network based on Elman, and they proved the accuracy of the Elman neural network in short-term predictions. In addition, the research shows that a genetic algorithm (GA) can effectively improve the training speed and accuracy of the neural network by optimizing Elman's connection weight and threshold.

In this study, the Shuizhuyuan landslide in the Three Gorges Reservoir area was taken as a case study. First, VMD was used to decompose the landslide displacement sequence into three subsequences: trend displacement, periodic displacement, and random displacement. Combined with the gray relational analysis method, the influencing factors that affect landslide displacement deformation were selected. Then, the training set, the test set, and the validation set were divided in the dataset. The GA–Elman model was used to predict and analyze the training set and the test set to determine the optimal training combination model. Finally, the displacement prediction of the landslide's cumulative displacement validation set was realized on the basis of the optimal prediction model. The main contributions of this paper are as follows: (1) the displacement data and the influencing factor data are decomposed by VMD and respectively integrated into the trend displacement dataset, the periodic displacement dataset, and the random displacement dataset; (2) each dataset is substituted into the GA–Elman model to train the model and make predictions; (3) the effect of model prediction is compared with the evaluation index to obtain the optimal prediction model; (4) the validation dataset is substituted to verify the prediction effect of the optimal prediction model.

## 2. Theory and Method

### 2.1. Variational Mode Decomposition

Variational modal decomposition was first proposed by K. Dragomiretskiy and D. Zosso in 2014 [40]. It is an adaptive, completely non-recursive approach to modal variational problem and signal processing. The basis of this method is to construct a variational problem and then solve the variational problem by decomposing a deterministic real-valued signal $Y$ into a discrete number of modes $Y_k(t)$, $k = 1, 2, 3, \cdots, K$. In this process, it is assumed that each decomposed mode fluctuates around a central pulsation. The biggest advantage of VMD compared with EMD is that it can determine the number of modal decompositions by itself. It overcomes the problems of endpoint effects and modal component aliasing in EMD, and it has a solid theoretical foundation. It can be used to deal with nonlinear sequences with poor regularity and high complexity, and to decompose nonlinear sequences into relatively stable subsequences.

The process whereby VMD decomposes displacement data is as follows:

Step 1: Construct the variational problem, decompose the signal $Y$ into $K$ components, determine the penalty parameter $\alpha$ and the rising step $\tau$, and set the constraint condition that the sum of all modes is equal to the original signal. The corresponding constraint variational expression is

$$\begin{cases} \min\limits_{\{u_k\}\{\omega_k\}} \left\{ \sum\limits_{k=1}^{K} \left\| \partial_t \left[ \left( \sigma(t) + \frac{j}{\pi t} \right) u_k(t) \right] e^{-j\omega_k t} \right\|_2^2 \right\} \\ s.t. \sum\limits_{k=1}^{K} u_k = f(t) \end{cases} \tag{1}$$

where $\{u_k\}$ and $\{\omega_k\}$ are the decomposed modal component and the center frequency of the modal component, respectively, $\left(\sigma(t) + \frac{j}{\pi t}\right) u_k(t)$ is the analytical signal of the modal component, and $f(t)$ is the original signal.

Step 2: By introducing the Lagrange penalty operator $\lambda$, the constrained problem is transformed into an unconstrained problem, and Equation (2) is obtained.

$$
\begin{aligned}
L(\{u_k\}, \{\omega_k\}, \lambda) = {} & \alpha \sum_k \left\| \partial_t \left[ \left( \sigma_t + \frac{j}{\pi t} \right) u_k(t) \right] e^{-j\omega_k t} \right\|_2^2 \\
& + \left\| f(t) - \sum_k u_k(t) \right\|_2^2 + \left\langle \lambda(t), f(t) - \sum_k u_k(t) \right\rangle
\end{aligned}
\tag{2}
$$

where $\alpha$ is the penalty factor.

Step 3: Substitute the saddle point obtained in the second step into the first step, and the value obtained is the solution of the model. The detailed process of alternating the iterative optimization of $\{u_k\}$, $\{\omega_k\}$, and $\lambda$ when solving with VMD is shown in Equations (3)–(5).

$$
\hat{u}_k^{n+1}(\omega) \leftarrow \frac{\hat{f}(\omega) - \sum_{i \neq k} \hat{u}_i(\omega) + \hat{\lambda}(\omega)/2}{1 + 2\alpha(\omega - \omega_k)^2}
\tag{3}
$$

$$
\omega_k^{n+1} \leftarrow \frac{\int_0^\infty \omega \left| \hat{u}_k^{n+1}(\omega) \right|^2 d\omega}{\int_0^\infty \left| \hat{u}_k^{n+1}(\omega) \right|^2 d\omega}
\tag{4}
$$

$$
\hat{\lambda}^{n+1}(\omega) \leftarrow \hat{\lambda}^n(\omega) + \gamma \left( \hat{f}(\omega) - \sum_k \hat{u}_k^{n+1}(\omega) \right)
\tag{5}
$$

The $K$ IMF components of the original signal $f(t)$ can be obtained through the above formula.

### 2.2. Elman Neural Network

The Elman neural network is a neural network with a local recursive function [22]. Compared with traditional neural networks, the Elman neural network introduces a fixed feedback link, which can monitor data changes in real time and enhance the network's ability to dynamically process information. The basic premise is to minimize the mean square error between the actual output and the expected output by using the least square method and the gradient search technique. Elman's main structure comprises two parts: the feedforward connection and the feedback connection. The feedforward connection includes the input layer, the hidden layer, and the output layer. The input layer unit plays the role of the signal transmission, the output layer unit plays a linear weighting role, and the transfer function of the hidden layer unit can adopt a tansig nonlinear function. The feedback connection is composed of a group of "connection" units. Its function in the network is to remember the output value of the hidden layer unit at the previous instance, and return it to the input of the network. It can be regarded as a one-step delay operator. Therefore, the biggest advantage of the Elman neural network is its memory ability, which can better reflect the nonlinear and dynamic characteristics of the model.

### 2.3. GA–Elman Model

GA has three main applications in the field of neural networks: training connection weights, designing network structures, and finding optimal learning rules. The first two applications have been widely studied by many scholars and have achieved good results. However, most published studies focus on feedforward neural networks rather than recursive neural networks. In this study, we use GA to optimize the initial weight and threshold of the Elman neural network. The traditional Elman model adopts the gradient descent method when updating the weight. This method not only runs slowly, but

also easily falls into local optimization, which cannot ensure the accuracy of the model's prediction results [41]. The GA's global search for the optimal solution can make up for this deficiency of the Elman neural network [42]. Using the data optimized by the GA as the weight and threshold of the Elman neural network input can greatly improve the accuracy of the model [43].

### 2.4. Displacement Prediction Process

The landslide displacement prediction process based on VMD and the GA–Elman model used in this paper is shown in Figure 1. The main prediction steps are described below.

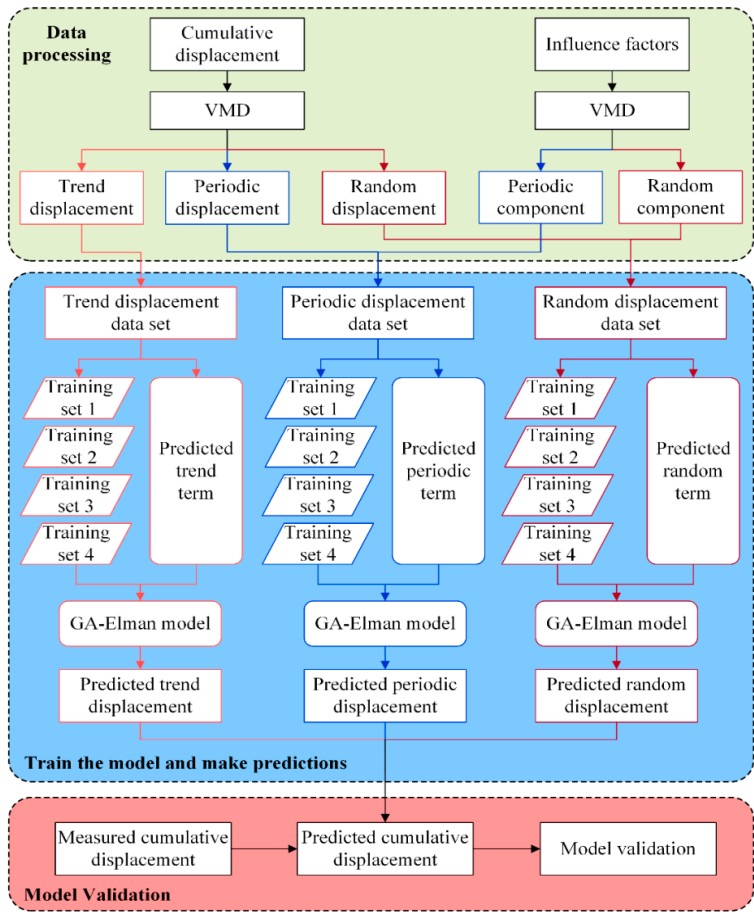

**Figure 1.** Flowchart of displacement prediction.

(1)  Preprocess the monitoring data. The displacement, rainfall, and reservoir level data observed at the monitoring points are preprocessed, and the types of influencing factors are identified.

(2)  Decompose the data. VMD is used to decompose the displacement data into three subseries of trend, periodic, and random terms, and to decompose the influence factor data into two subseries of periodic and random terms.

(3)  Consolidate the datasets. The decomposed data are integrated into corresponding datasets according to the decomposition; then, the training set, test set, and validation set are divided up.

(4)  Train the GA–Elman model and compare the prediction results. The training set is substituted into the GA–Elman model separately to train the model parameters, and then the test set is substituted into the model to determine the optimal training combination.

(5)   Determine the optimal prediction model for cumulative displacement. The optimal prediction model for cumulative displacement is obtained by accumulating the optimal training combinations from the trend dataset, the periodic dataset, and the random dataset.

(6)   Verify the feasibility of the optimal prediction model. Substitute the validation set data into the optimal prediction model and verify the feasibility of the model combined with the operation results of each evaluation index.

## 3. Research Area

### 3.1. General State of the Engineering Geology of the Shuizhuyuan Landslide

The Shuizhuyuan landslide was located in Group 1 of Ganju Village, Quchi Township, Wushan County, on the left bank of the Yangtze River (31°00′59.37″ E, 109°43′27.33″ N). It was 14.82 km away from the new urban area of Wushan and 170 km away from the Three Gorges Dam in the east (Figure 2).

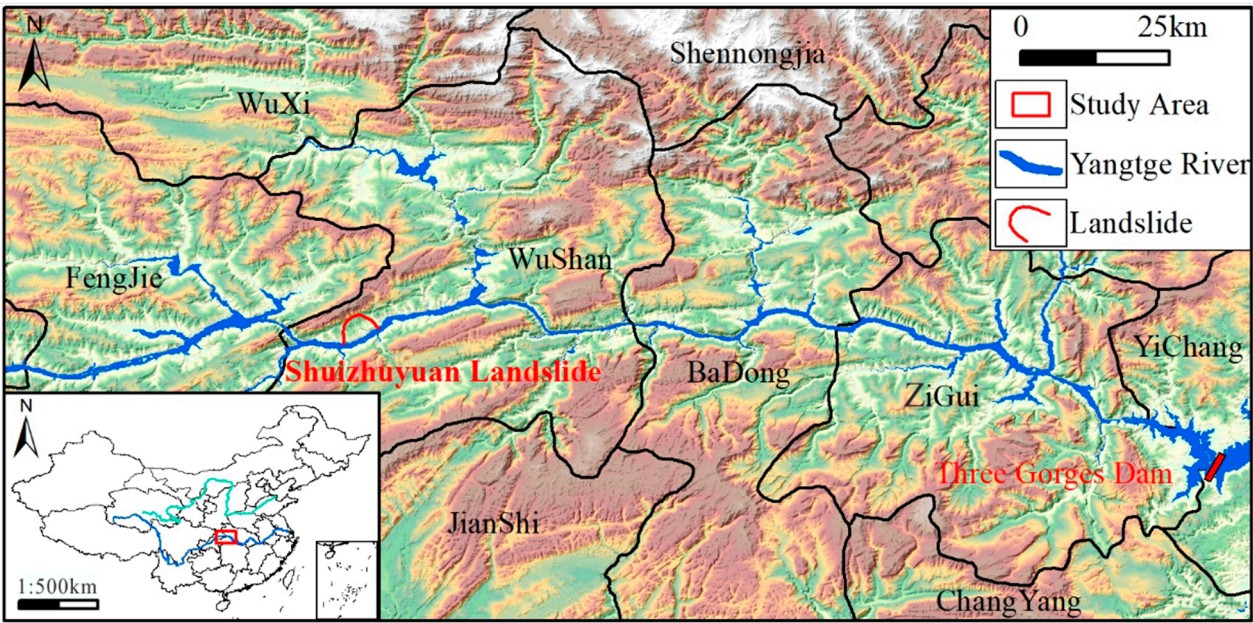

**Figure 2.** Location of the Shuizhuyuan landslide.

The elevation of the rear edge of the landslide was 350 m and the elevation of the front edge was 90 m; the front edge was submerged below the water level of the reservoir area at 145 m. The relative elevation difference was 260 m, and the main sliding direction was 150°. The landslide had a length of about 800 m, a width of 360–1200 m, an average thickness of 30 m, an area of $62 \times 10^4$ m$^2$, and a volume of $1850 \times 10^4$ m$^3$. The landslide potentially threatened 375 people in 65 households (191 permanent residents) and the safety of shipping in the Yangtze River. In 2002, it was found that there was a crack, 300 m long and 2–3 cm wide, in the middle and rear edge of the landslide. In May 2003, it was first found that the upper house (adobe house) had signs of cracking and deformation, with a maximum crack size of 20 cm. Since the monitoring work was carried out in 2006, the landslide deformation has generally shown a nearly uniform creep deformation trend. The main deformation characteristic is that the deformation of the front edge of the landslide was large, and the deformation of the rear edge was relatively small. Small collapses occurred at the wading part of the front edge of the landslide, showing obvious deformation characteristics of a traction landslide.

The sliding mass was composed of loose Quaternary deposits. The engineering geological profile of the landslide is shown in Figure 3. The bedrock outcrop section in the middle–lower part of the landslide comprised siltstone and silty mudstone of the Middle

Triassic Badong Formation ($T_2b$). The exposed stratum in the middle and trailing edge of the landslide was Upper Triassic Xujiahe Formation ($T_3xj$) sandstone. The sliding zone material of the landslide was mainly silty clay, due to the sliding zone being filled with water and softer plastic, which are prone to sliding deformation. On the basis of the above analysis, the landslide could be characterized as a soil landslide with precipitation and reservoir as the main inducing factors.

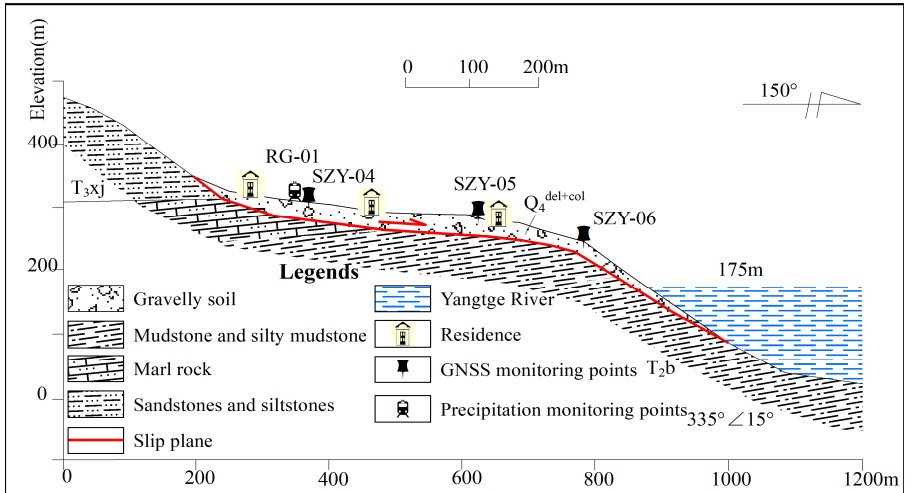

**Figure 3.** The engineering geological profile of the Shuizhuyuan landslide.

Therefore, the main aspects of the landslide that require monitoring are surface displacement, rainfall, change in the reservoir water level, and macro patrol monitoring. Seven GNSS deformation monitoring points and one GNSS reference point were identified on the landslide, forming three longitudinal monitoring profiles, one rainfall monitoring point, and one point for monitoring reservoir water level change using the collected data. The layout plan of the monitoring network is shown in Figure 4. Since the deployment of the monitoring equipment, a continuous state of deformation has been observed in the landslide; the deformation of the middle front edge on the left side of the landslide was relatively strong. The Shuizhuyuan landslide is a typical landslide located in the Three Gorges Reservoir area. It is of great significance to analyze it and accurately predict its development.

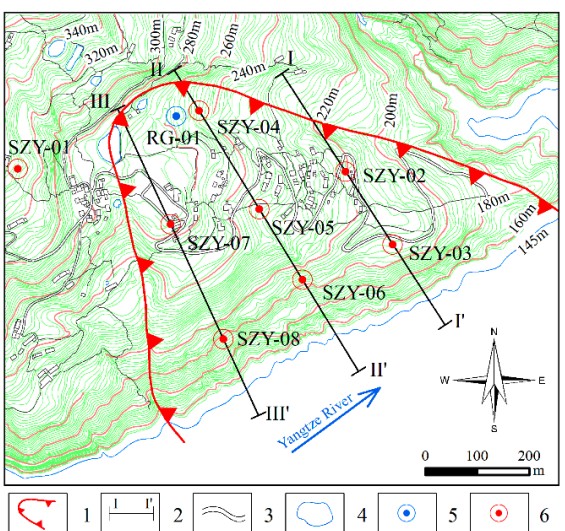

1.Landslide boundaries  2.Monitoring profile  3.Road  4.Pond
5.Preciptitation moitoring points  6.Gnss monitoring points

**Figure 4.** Layout plan of the Shuizhuyuan landslide monitoring network.

### *3.2. Landslide Monitoring Data Preprocessing*

This paper selected the No. 3 GNSS monitoring station, which recorded obvious deformation of the Shuizhuyuan landslide in the Three Gorges Reservoir area, for data analysis. We used as the research object the data produced at the observation point across a period of 840 days from 1 July 2018 to 18 October 2020. The data were processed in terms of weeks, and a total of 120 weeks of data were obtained (Figure 5). The data for the first 80 weeks were used as the training set used to train the model and determine its relevant parameters. The data from 81 to 100 weeks were used as the test set of the model to compare the results of each model and determine the optimal prediction model. The data from 101 to 120 weeks were used as the validation set of the model to compare the error between the prediction results of the optimal model and the actual displacement, in order to verify the feasibility of the model. To determine the impact of different combinations of training sets on the prediction accuracy of the model, and to analyze the timeliness of the displacement data, four groups of different training sets were constructed to train the model, as shown in Table 1. The test set took the data from the middle 20 weeks, i.e., the data from 81 to 100 weeks. The optimal training combination was determined by comparing the operation results of different combinations of training sets input into the GA–Elman model.

**Table 1.** The training set of the prediction model.

| Model Number | Training Set Data Volume | Time Included in the Training Data | | | |
|---|---|---|---|---|---|
| | | 1–20 Weeks | 21–40 Weeks | 41–60 Weeks | 61–80 Weeks |
| Model 1 | 20 | | | | √ |
| Model 2 | 40 | | | √ | √ |
| Model 3 | 60 | | √ | √ | √ |
| Model 4 | 80 | √ | √ | √ | √ |

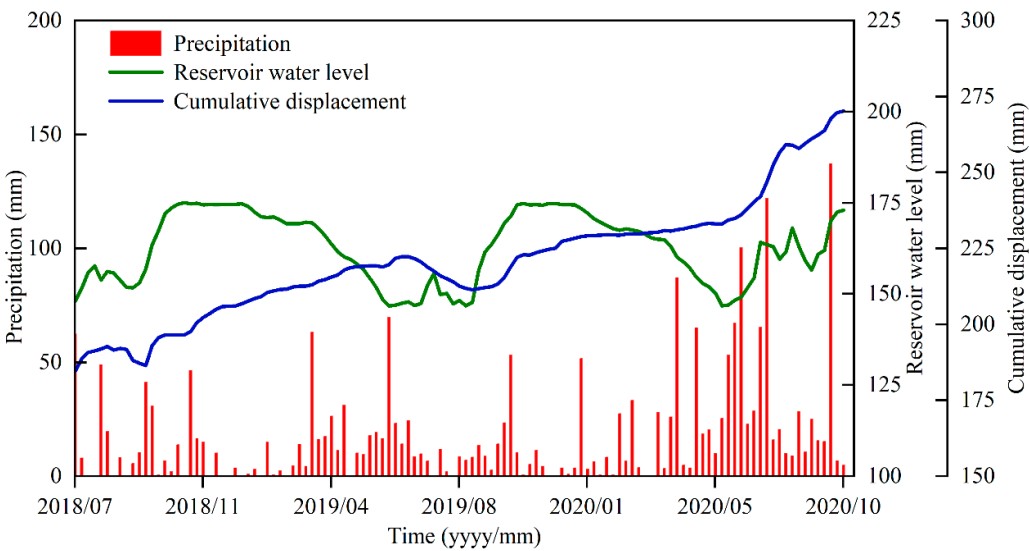

**Figure 5.** Precipitation, reservoir water level, and cumulative displacement at the SZY-03 monitoring point of the Shuizhuyuan landslide.

## 4. Application Research and Method Comparison

### *4.1. Monitoring Data Processing*

The monitoring data obtained from the Shuizhuyuan landslide mainly comprised GNSS surface displacement data, precipitation data, and reservoir water level data. In this section, we carry out the following steps: (i) decomposing the landslide displacement data and the influencing factor data into different components using VMD; (ii) conducting

gray correlation analysis on the decomposed displacement and influencing factor data, according to the number of data contained in different sets of training sets.

### 4.1.1. Decomposition of Landslide Displacement Data

The displacement deformation of a landslide is influenced by multiple factors. Therefore, in order to obtain an accurate prediction result of the landslide displacement, it is necessary to decompose the acquired landslide displacement into several subcomponents, corresponding to the influences of different factors [11,23,44,45]. The landslide undergoes displacement deformation under the influence of its own geological conditions. This displacement deformation component caused by the landslide itself is defined as the trend displacement component. Many studies have shown that seasonal precipitation and changes in the water level of the reservoir area in the Three Gorges region indirectly affect the displacement deformation of the landslide [14,32,46]. Therefore, the displacement deformation components caused by the influencing factors with periodic changes are defined as the periodic displacement components. In addition, the landslide displacement is deformed by other external factors, such as artificial activities and earthquakes. These kinds of displacement deformation components are defined as random displacement components.

The parameters need to be set before VMD can be used for the cumulative 120 weeks of displacement data. Considering that the displacement component contains at least three terms (trend, period, and random), the modal number K = 3. During the experiment, the penalty parameter and the rising step have a great influence on the decomposition results of the data, especially in the periodic term and random term. The parameters were set to $\alpha = 2000$, $\tau = 0$, DC = 0, and init = 1 after several trials.

It was determined that, when K = 3, the decomposed periodic term was not obvious. Therefore, K = 4 was adopted for the decomposition, and the results of the decomposition were processed analytically. The more significant IMF1 and IMF2 were used in the trend term displacement. IMF3, which had good periodicity and low-frequency characteristics, was used as the periodic term. The higher-frequency IMF4 was used as the random term. The decomposition results are shown in Figure 6.

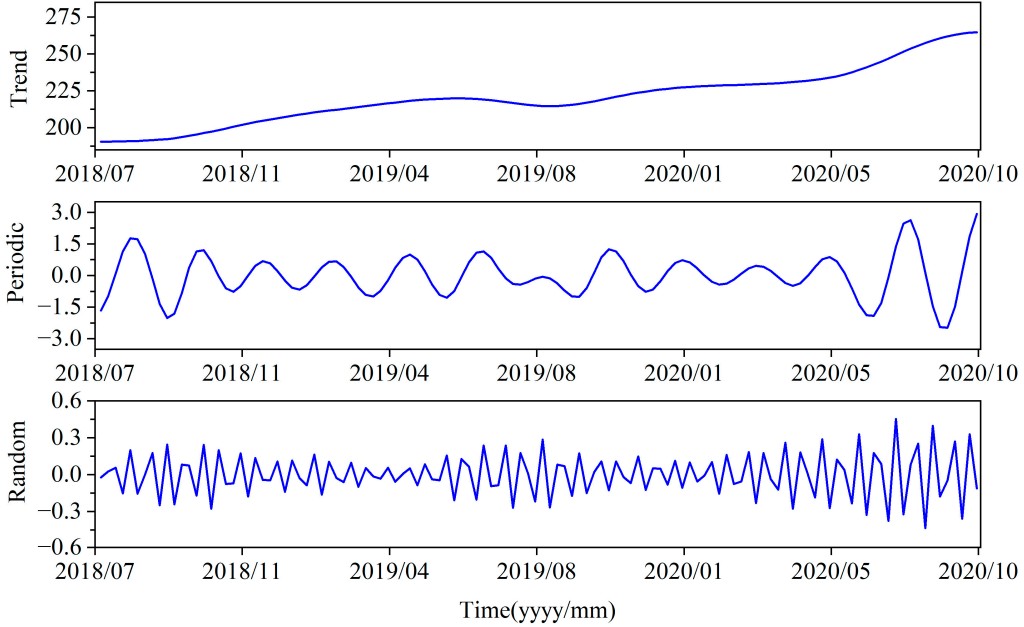

**Figure 6.** Decomposition results of the displacement data after processing.

### 4.1.2. Selection and Decomposition of Influencing Factors

The trend displacement of the landslide is related to the internal factors of the landslide. The rock and soil composition of the landslide, changes in the internal stress, and changes in

geometric shape, along with other factors, influence the trend displacement of the landslide; moreover, these factors often change over time. Therefore, the main factor influencing the trend displacement of the landslide is the monitoring time. It can be seen from Figure 5 that the cumulative displacement increased to varying degrees with the amount of precipitation. A large increase in cumulative displacement occurred when precipitation was abundant, and displacement slowed down when precipitation was relatively low, which shows that the cumulative displacement increased more obviously in the rainy season. In addition, according to related research [47], the reservoir water level also has obvious periodicity. When the water level in the reservoir area tends to fall, the water pressure applied to the surface of the slope decreases and the support capacity of the slope is weakened, accelerating the deformation of the landslide; at these times, the accumulated displacement of the landslide appears to rise rapidly. When the water level in the reservoir area rises, the water pressure acting on the surface of the slope is greater than the force of the slope sliding outwards, inhibiting landslide deformation and slowing down the increase in cumulative displacement. The above analysis demonstrates that precipitation and reservoir levels are important external factors that influence landslide deformation.

On the basis of previous research [38], four factors were selected as those influencing periodic displacement: the displacement increment last week (P1), the cumulative precipitation this week (P2), the average elevation of the reservoir water level this week (P3), and the variation range of the reservoir water level this week (P4). For the VMD decomposition of influencing factor data, the decomposition mode number $K$ was set first. Considering that there was no possibility of trend items being present in the list of influencing factors, K = 2 was set during the decomposition. The decomposed component with a large proportion and low frequency was taken as the periodic term component of the influencing factors. The decomposed component with a relatively small proportion and high frequency was taken as the random term component of the influencing factors. The effect of decomposing the influencing factors is shown in Figure 7.

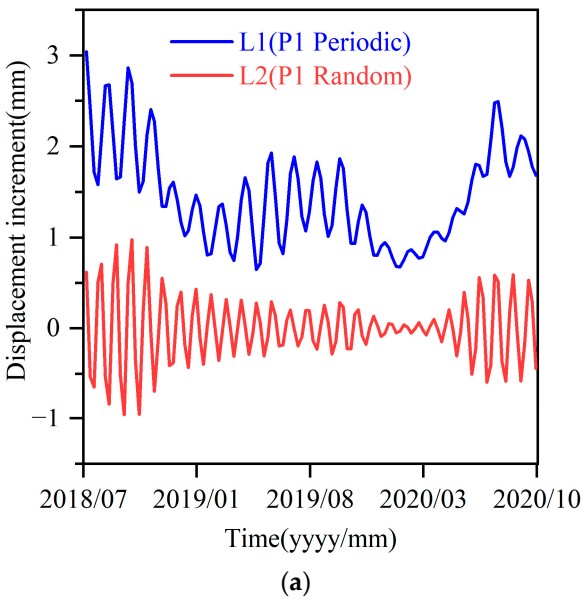
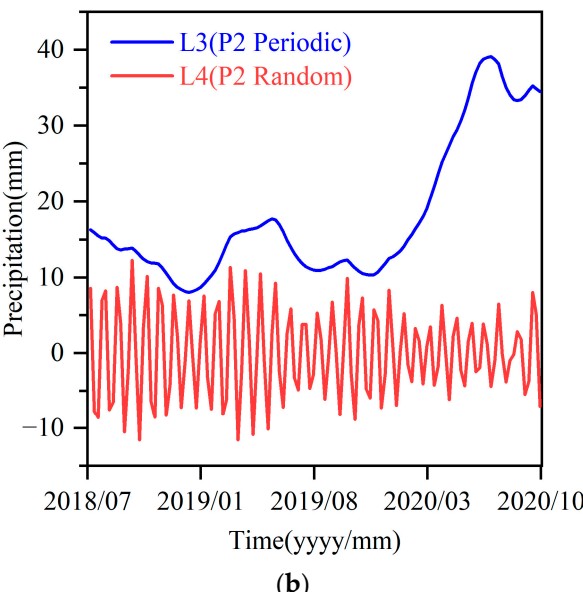

(**a**)                                      (**b**)

**Figure 7.** *Cont.*

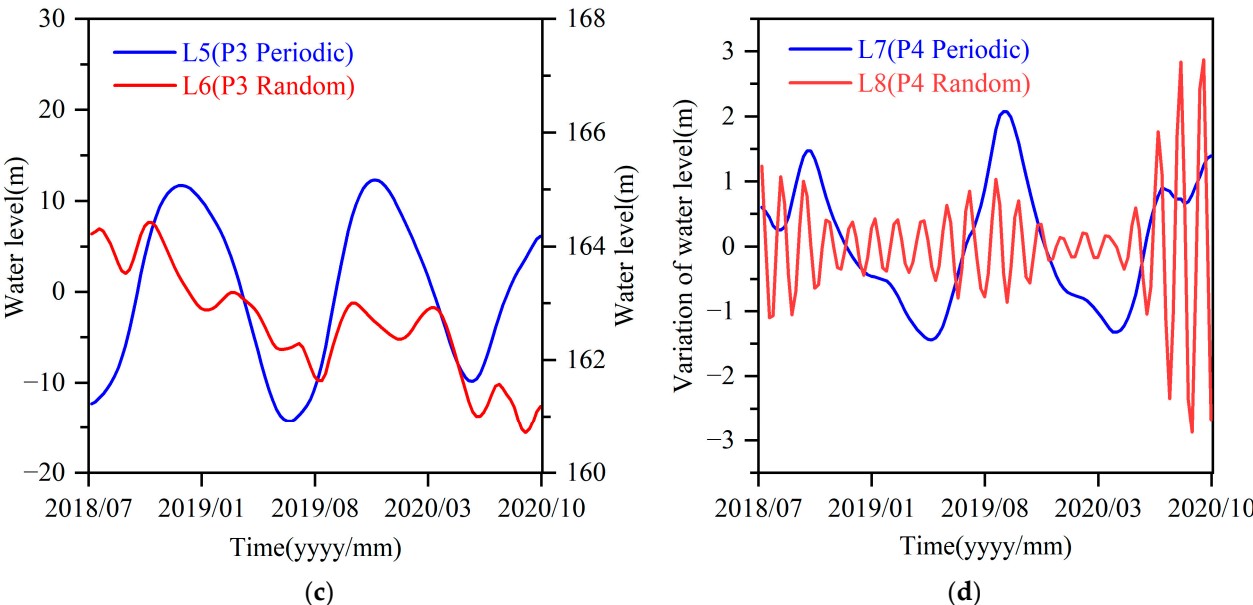

**Figure 7.** Decomposition results of the influencing factors: (**a**) decomposition result of P1; (**b**) decomposition result of P2; (**c**) decomposition result of P3; (**d**) decomposition result of P4.

### 4.1.3. Relational Analysis of Displacement Components and Influencing Factor Components

In order to verify the rationality of the influencing factor selection after decomposing the displacement data and the influencing factor data, it was necessary to analyze the correlation between the displacement data components and the influencing factor components. Gray relational degree theory is an evaluation method in gray system theory [48]. It can compare the relational degrees between different sequences and is suitable for studying the problems of insufficient data, poor information, and uncertainty [49]. The gray relational degree model holds that, when the resolution coefficient $\rho$ is 0.5, the closer the value of the correlation degree $r$ is to 1, and the better the correlation is between the two variables. In order to ensure the effectiveness and reliability of the data selection, gray relational analysis was used for the displacement component data and the influencing factor component data when the number of training sets was 20 weeks, 40 weeks, 60 weeks, and 80 weeks. The results are shown in Table 2. When the data volume of the training set was 40 weeks, the data in the periodic component dataset and the random component dataset both showed a strong correlation. It can also be seen from the table that the degree of correlation between the displacement component data and the influencing factor data had little to do with the type of influencing factor, but was mainly related to the number of weeks of the training set contained in the dataset; this proves that the selected influencing factors had a strong correlation with the displacement data.

**Table 2.** Analysis of the relational degree between the displacement component and the influence factor component.

| Model Number | Component Type | Influencing Factors | | | |
|---|---|---|---|---|---|
| | | P1 | P2 | P3 | P4 |
| Model 1 | Periodic component | 0.5769 | 0.5751 | 0.5747 | 0.5734 |
| | Random component | 0.5663 | 0.5693 | 0.5690 | 0.5685 |
| Model 2 | Periodic component | 0.8505 | 0.8509 | 0.8507 | 0.5349 |
| | Random component | 0.9759 | 0.5920 | 0.9762 | 0.9764 |
| Model 3 | Periodic component | 0.6532 | 0.6559 | 0.6544 | 0.6703 |
| | Random component | 0.8916 | 0.6057 | 0.9194 | 0.9139 |
| Model 4 | Periodic component | 0.6293 | 0.6300 | 0.6297 | 0.6357 |
| | Random component | 0.8234 | 0.8957 | 0.9099 | 0.6676 |

### 4.2. Prediction of Trend Displacement

When the trend component data of the displacement were substituted into the model for prediction, the model prediction results under different training sets were compared using evaluation indices such as MAPE (mean absolute percentage error), MSE (mean square error), RMSE (root-mean-square error), and $R^2$ (coefficient of determination). When the GA–Elman model is used for training and prediction, it is necessary to determine the values of the relevant parameters involved in the model. Through multiple trial calculations, the relevant parameters of GA were set as follows: the number of population iterations was 30, the population size was 30, the crossover probability was 0.3, and the mutation probability was 0.1. In the setting of relevant parameters for the Elman algorithm, the learning rate was 0.5, the momentum factor was 0.9, and the model accuracy was 0.00001.

The operation results, generated after the trend component dataset was substituted into the model, are shown in Figure 8a–d. It can be seen that, when the training set was 40 weeks, 60 weeks, and 80 weeks, there was a concave phenomenon of nearly 5 mm in one section of the trend displacement. By verifying the original monitoring data of the monitoring points, it was confirmed that there was a decrease in one section of the displacement monitoring data. Through further observation and analysis, no obvious changes in the precipitation data and the water level data for the reservoir area were identified during this period. On the basis of the situation detailed above, it can be suggested that the concave displacement of the trend term in this section was due to the influence of external environmental factors (search satellite, fog, moisture, etc.), which influenced the monitoring accuracy. However, this did not affect the overall working performance of the monitoring equipment. The results of the runs show that the prediction accuracy of the model tended to increase and then decrease with the increase in the base data in the training set. The relevant evaluation index values in the prediction results are shown in Table 3. According to the operation results in Table 3, the MAPE, MSE, RMSE, and $R^2$ of Model 2 were 0.06%, 0.0361, 0.9876, and 0.0217, respectively, which are better results than those produced by the other models. The prediction accuracy of this model was the highest. That is to say, in the prediction process of trend displacement, when the basic training set data volume was 40 weeks, the model prediction effect was the best. This indicates that the timeliness of the trend displacement component data of the landslide was strong, and the real-time monitoring of landslide displacement data must be strengthened.

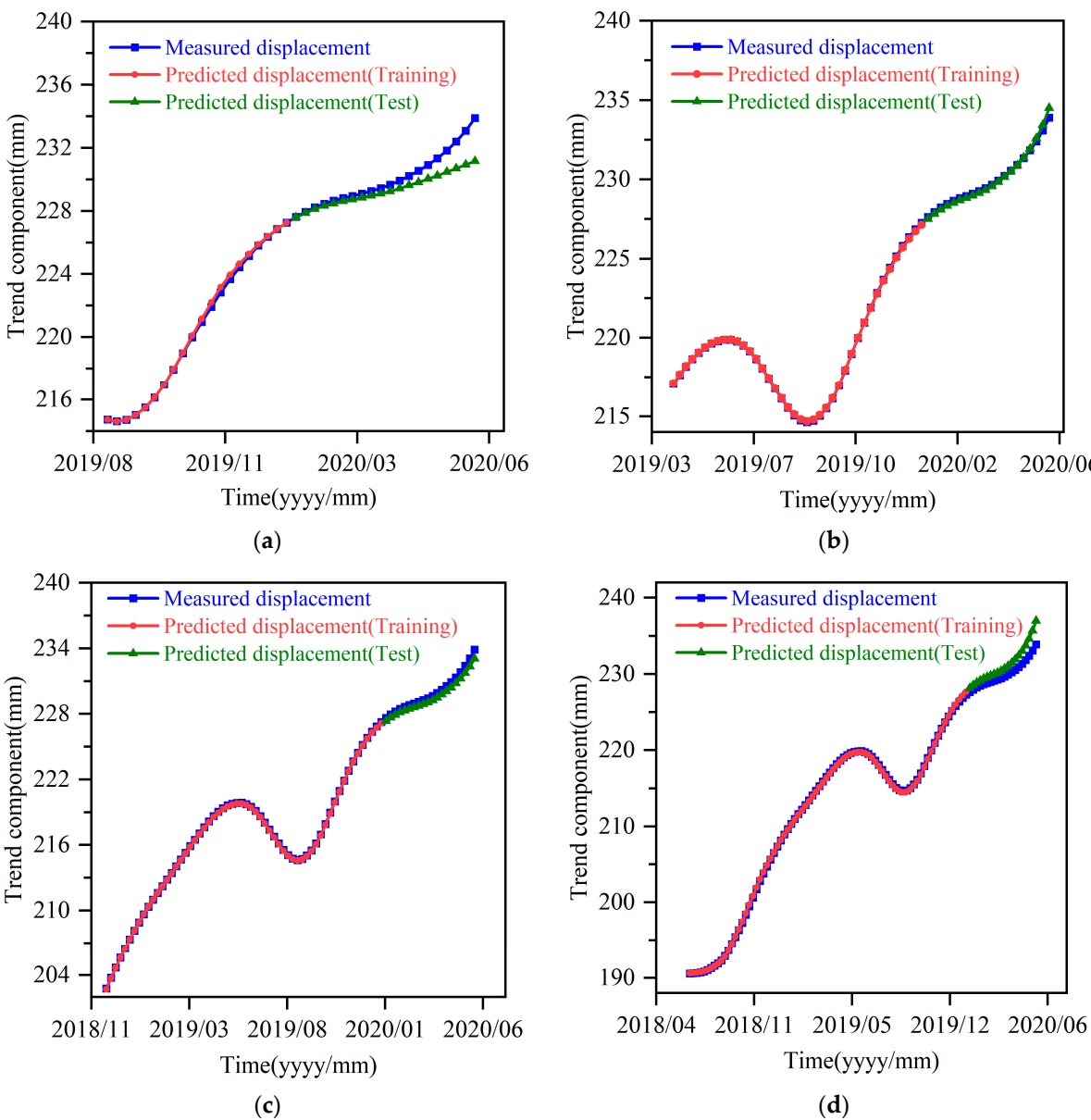

**Figure 8.** Trend displacement prediction results for different training sets: (**a**) prediction results of Model 1; (**b**) prediction results of Model 2; (**c**) prediction results of Model 3; (**d**) prediction results of Model 4.

**Table 3.** Evaluation index values of different models in trend displacement.

| Model Number | Evaluation Index | | | |
|:---:|:---:|:---:|:---:|:---:|
| | MAPE (%) | MSE | RMSE | $R^2$ |
| Model 1 | 0.3000 | 1.0276 | 0.6998 | 0.6462 |
| Model 2 | 0.0600 | 0.0361 | 0.0217 | 0.9876 |
| Model 3 | 0.2000 | 0.2279 | 0.4529 | 0.9215 |
| Model 4 | 0.5400 | 2.0526 | 1.2501 | 0.2933 |

### 4.3. Prediction of Periodic Displacement

The periodic term component dataset was substituted into the model, and the operation results are shown in Figure 9a–d. The evaluation index values are shown in Table 4. According to the values of various evaluation indices in Table 4, it can be seen that the prediction accuracy of the model showed a gradual decrease as the training set of data

increased. When the training set was 20 weeks of data, the best run for Model 1, MAPE was 1.5661%, MSE was 0.0001, RMSE was 0.0097, and $R^2$ was 0.9994. This means that the periodic displacement component data were also timely, and their timeliness was stronger than that of the trend item component.

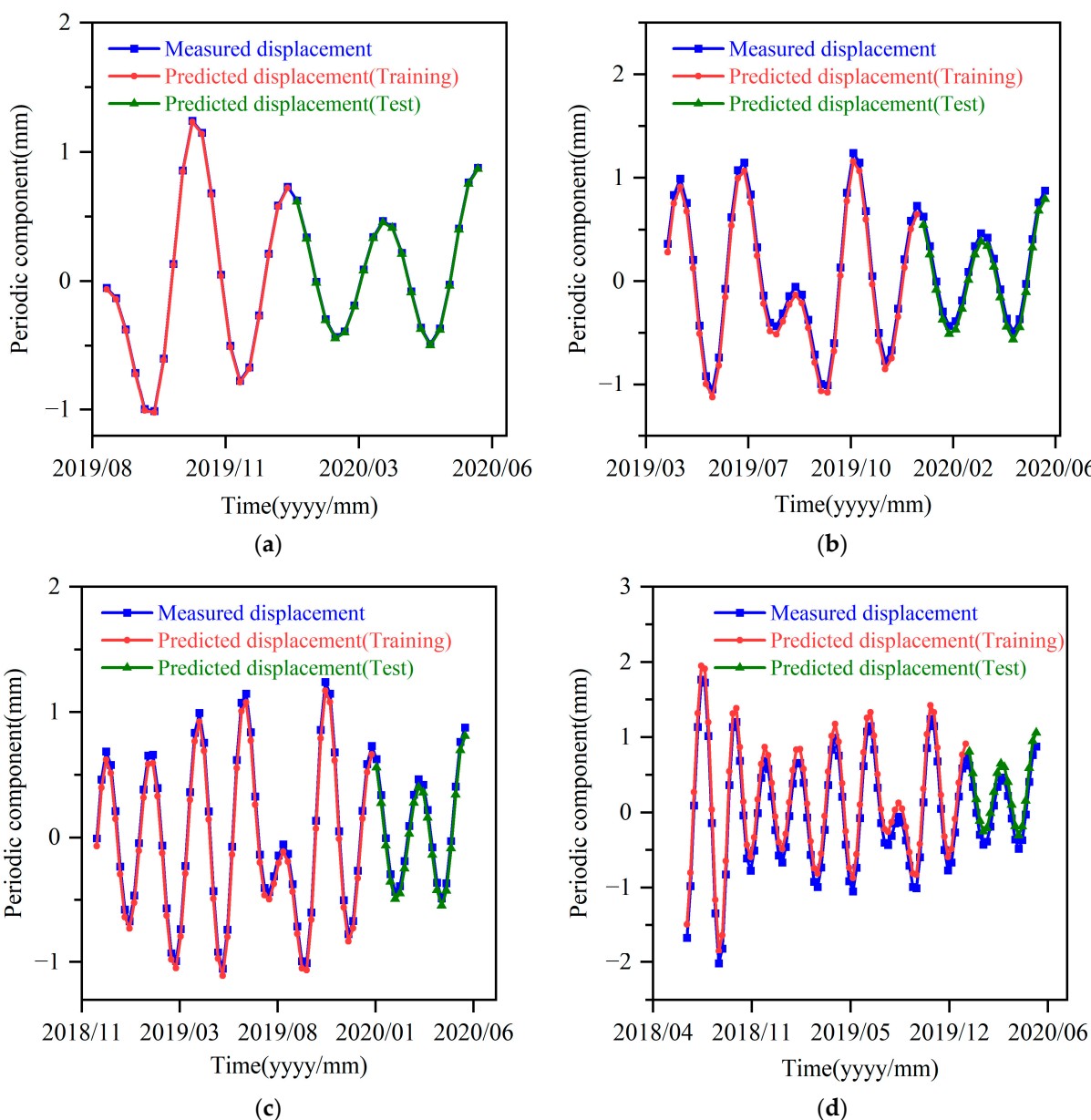

**Figure 9.** Periodic displacement prediction results of different training sets: (**a**) prediction results of Model 1; (**b**) prediction results of Model 2; (**c**) prediction results of Model 3; (**d**) prediction results of Model 4.

**Table 4.** Evaluation index values of different models of periodic displacement.

| Model Number | Evaluation Index | | | |
|---|---|---|---|---|
| | MAPE (%) | MSE | RMSE | $R^2$ |
| Model 1 | 1.5661 | 0.0001 | 0.0097 | 0.9994 |
| Model 2 | 10.5000 | 0.0066 | 0.0811 | 0.9611 |
| Model 3 | 7.9905 | 0.0039 | 0.0628 | 0.9766 |
| Model 4 | 23.3792 | 0.0327 | 0.1808 | 0.8067 |

### 4.4. Prediction of Random Displacement

After the random item component dataset was substituted into the model, and the operation results were generated, as shown in Figure 10a–d.

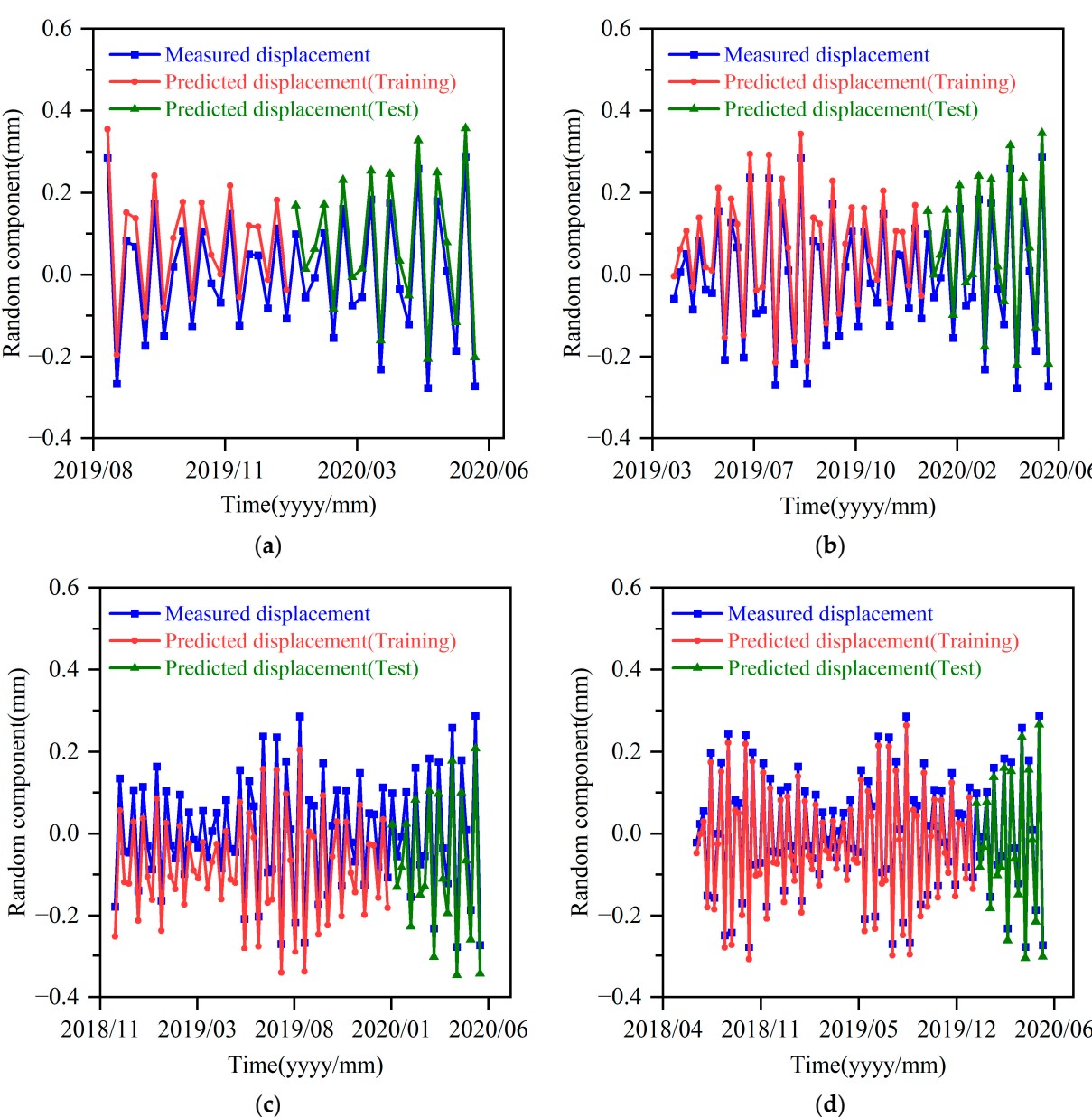

**Figure 10.** Random displacement prediction results of different training sets: (**a**) prediction results of Model 1; (**b**) prediction results of Model 2; (**c**) prediction results of Model 3; (**d**) prediction results of Model 4.

The evaluation indices in Table 5 show that, as the data in the training set increased, the model prediction accuracy went through a trend of increasing, then decreasing, and then increasing again. The best operation results were generated by Model 4, i.e., when the training set data volume was 80 weeks. Here, MAPE was 13.73%, MSE was 0.0007, RMSE was 0.0261, and $R^2$ was 0.9765. Moreover, the random displacement term accounted for a small proportion in the total displacement, and the influencing factors were difficult to identify.

**Table 5.** Evaluation index values of different models of random displacement.

| Model Number | Evaluation Index | | | |
|---|---|---|---|---|
| | MAPE (%) | MSE | RMSE | $R^2$ |
| Model 1 | 32.96 | 0.0049 | 0.0699 | 0.8326 |
| Model 2 | 112.88 | 0.0031 | 0.0560 | 0.8926 |
| Model 3 | 33.87 | 0.0057 | 0.0758 | 0.8028 |
| Model 4 | 13.73 | 0.0007 | 0.0261 | 0.9765 |

### 4.5. Prediction of Cumulative Displacement

The cumulative displacement prediction model could be obtained by accumulating the predicted values obtained by substituting the trend item, periodic item, and random item datasets into the model. The results are shown in Figure 11a–d.

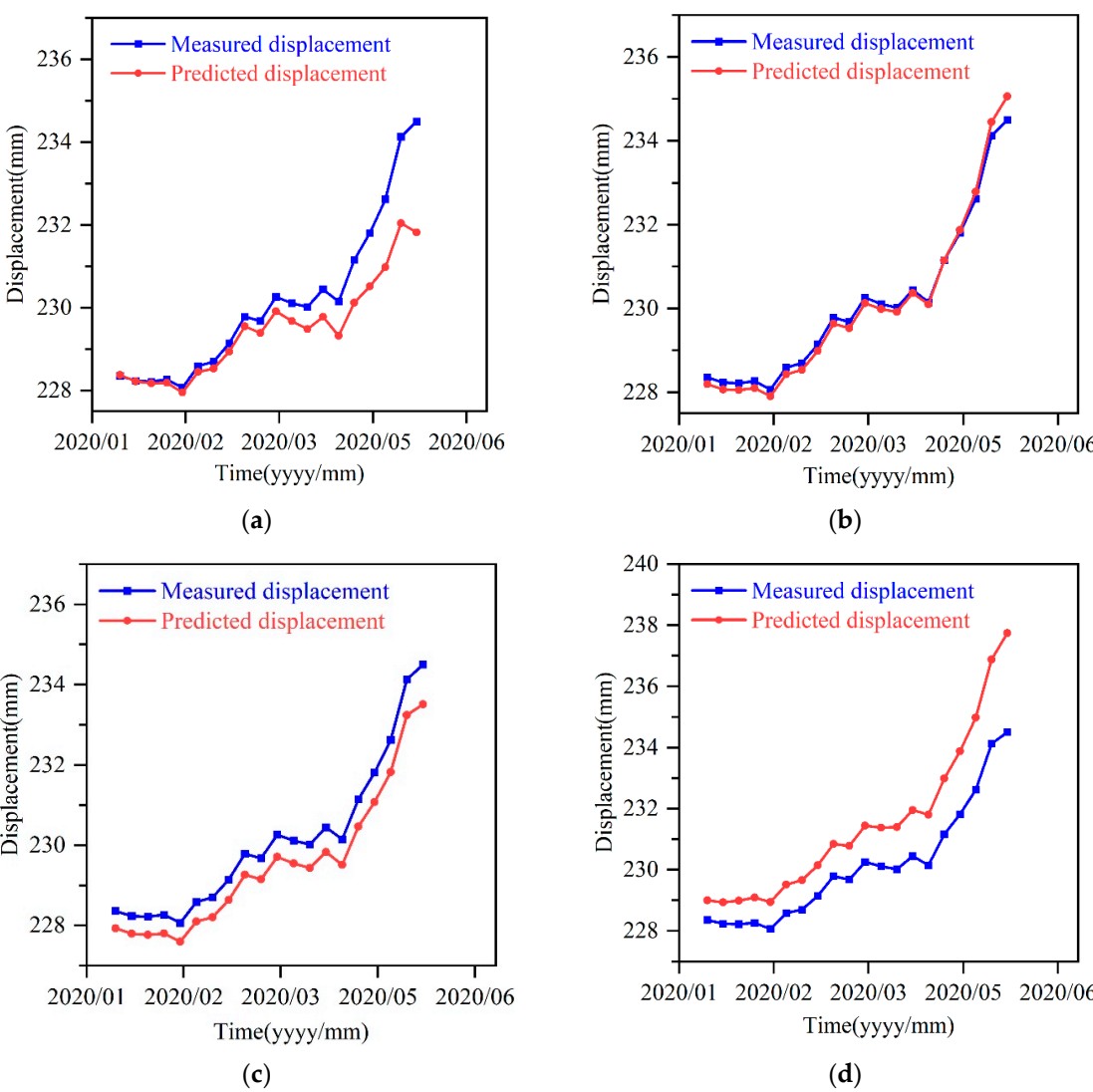

**Figure 11.** Cumulative displacement prediction results of different training sets: (**a**) prediction results of Model 1; (**b**) prediction results of Model 2; (**c**) prediction results of Model 3; (**d**) prediction results of Model 4.

The evaluation index values are shown in Table 6. From the graph, we can see more intuitively that the model fit best when the training set was 40 weeks, and the real value of the cumulative displacement was closest to the model prediction. The data in Table 6 show that the prediction effects of Model 2 were the best. At this time, MAPE was 0.1883%,

MSE was 0.0377, $R^2$ was 0.9891, and RMSE was 0.0469. However, in conjunction with the prediction results in Sections 4.2–4.4, the optimal prediction model for the trend component was Model 2, the optimal prediction model for the periodic component was Model 1, and the optimal prediction model for the random component was Model 4. Although the prediction effect of Model 2 was optimal on a whole, it was not locally optimal in all places, and there was room for improvement. After combining the optimal models corresponding to the trend term, periodic term, and random term, the test set was predicted. The results are shown in Figure 12, and the value of each evaluation index is shown in the last row of Table 6. Compared with Model 2, the evaluation index values tended to be more optimal.

**Table 6.** Evaluation index values of different models for cumulative displacement.

| Model Number | Evaluation Index | | | |
|---|---|---|---|---|
| | MAPE (%) | MSE | RMSE | $R^2$ |
| Model 1 | 0.2763 | 0.9469 | 0.6396 | 0.7261 |
| Model 2 | 0.1883 | 0.0377 | 0.0469 | 0.9891 |
| Model 3 | 0.2565 | 0.3728 | 0.5914 | 0.8922 |
| Model 4 | 0.6082 | 2.4631 | 1.4048 | 0.2875 |
| Combined model | 0.1685 | 0.0371 | 0.0384 | 0.9893 |

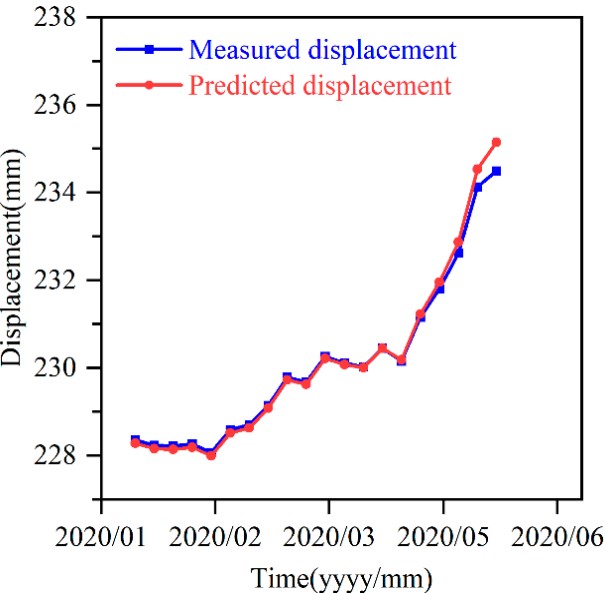

**Figure 12.** Cumulative displacement prediction results of the combined model.

*4.6. Feasibility Verification of the Prediction Model*

To further verify the prediction effect of the optimal combination prediction model obtained in Section 4.5, the data from 101 to 120 weeks were used as the validation set and substituted into the model for prediction. The prediction results are shown in Figure 13.

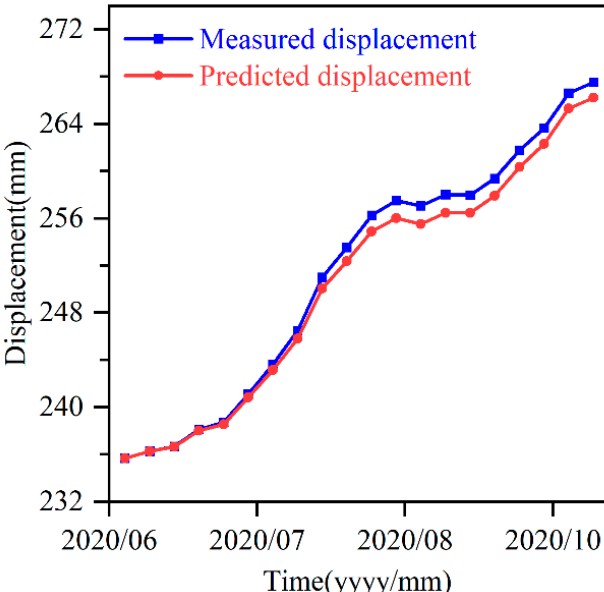

**Figure 13.** The cumulative displacement prediction results of the validation set.

It can be observed from Figure 13 that the two curves produced by the actual value of the validation set and the predicted value of the model were relatively close. The corresponding evaluation index values were MAPE = 0.3493%, MSE = 1.1635, RMSE = 0.9001, and $R^2$ = 0.9895. This further verifies that the combined model had a good prediction effect, thus offering a solution problem in the field of landslide displacement prediction.

*4.7. Model Comparison*

After determining the prediction model for the Shuizhuyuan area in the Three Gorges, this paper further compares the results of this model with other landslide displacement prediction models. In order to verify the validity and feasibility of VMD, the results of the model are compared with the results of the GA–Elman model, and, to verify the practicability of the GA–Elman algorithm, the model is compared with the Elman model. The comparison results are shown in Table 7. It can be seen that the MAPE values of the Elman and GA–Elman models were both greater than 100%; the prediction effect of the model was poor, and not as good as the prediction effect of the combined model. The model proposed in this paper had the advantage of higher prediction accuracy.

**Table 7.** Comparison of the prediction accuracy of different prediction models.

| Model Name | Evaluation Index | | | |
|---|---|---|---|---|
| | MAPE (%) | MSE | RMSE | $R^2$ |
| Elman | 372.55 | 98.6357 | 9.3709 | 0.3633 |
| GA–Elman | 153.04 | 23.1281 | 3.8904 | 0.8507 |
| VMD–GA–Elman (Combined model) | 0.3493 | 1.1635 | 0.9001 | 0.9895 |

**5. Discussion and Conclusions**

In this study, a new landslide displacement prediction method was proposed by combining the VMD, GA, and Elman models. Taking the Shuizhuyuan landslide in the Three Gorges Reservoir area as a case study, the trend, periodic, and random term components of the cumulative landslide displacement were obtained separately using VMD. The GA–Elman model was used to conduct an accurate prediction of landslide displacement, and the optimal combination model under different training sets was obtained. At the same time, the applicability of the model was verified in comparison with the basic Elman and

GA–Elman models. However, due to the classification of different training sets, some training sets exhibited an overfitting phenomenon in the process of prediction. Therefore, in the future landslide displacement prediction, the type of training set and the influencing factors of landslides should be selected reasonably to avoid the overfitting phenomenon and obtain the best landslide displacement prediction effect.

(1) In this paper, VMD was used to achieve the effective decomposition of landslide displacements, solving the modal mixing problem of traditional empirical modal decomposition. The Elman neural network was optimized using GA, which effectively solved the problem posed by the difficulty of determining the weights, thresholds, and neurons of the Elman neural network; moreover, it effectively improved the model's prediction accuracy.

(2) This study accounted for the internal and external factors that influence landslide deformation, such as past cumulative displacement, precipitation, and the reservoir water level. The changes in monitoring data were analyzed in detail, in conjunction with previous research, and four influencing factors were ultimately identified. The gray correlation among these four influencing factors and the displacement of the fluctuating term was greater than 0.5, indicating that the influencing factors were selected effectively.

(3) The prediction results showed that the model had high prediction accuracy and prediction capabilities with the effective acquisition of early monitoring data of landslides. This study, therefore, provides a new basis for predictions in the study of similar landslides.

**Author Contributions:** Conceptualization, C.W. and K.Y.; methodology, W.G., Q.M., X.W. and Z.Z.; software, W.G., Q.M. and Z.Z.; validation, W.G., C.W. and K.Y.; formal analysis, W.G. and C.W.; investigation, K.Y.; resources, W.G., Q.M. and X.W.; data curation, C.W. and K.Y.; writing—original draft preparation, W.G., Q.M., K.Y. and C.W.; writing—review and editing, W.G., Q.M., K.Y. and C.W.; visualization, Q.M. and Z.Z.; supervision, C.W.; project administration, K.Y. and W.G.; funding acquisition, K.Y. and C.W. All authors read and agreed to the published version of the manuscript.

**Funding:** This research was funded by the National Key Research and Development Program of China (No. 2019YFC150960403 and No. 2019YFC150960101) and the Open Project of the Technology Innovation Center for Geological Environment Monitoring of MNR (No. 2022KFK1212005).

**Institutional Review Board Statement:** Not applicable for studies not involving humans or animals.

**Informed Consent Statement:** Not applicable for studies not involving humans.

**Data Availability Statement:** The data presented in this study are available on request from the corresponding author. The data are not publicly available due to funding project confidentiality requirements.

**Acknowledgments:** The authors thank the Three Gorges Reservoir Geological Hazards—Chongqing Wushan Field Scientific Observation and Research Station of the Ministry of Natural Resources for providing the monitoring data of the Shuizhuyuan landslide.

**Conflicts of Interest:** The authors declare no conflict of interest. The funders had no role in the design of the study; in the collection, analyses, or interpretation of data; in the writing of the manuscript; or in the decision to publish the results.

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
