# Peer review of "Landslide Displacement Prediction Based on Variational Mode Decomposition and GA–Elman Model"

_applsci, doi:10.3390/app13010450_

Round 1

Reviewer 1 Report

The manuscript entitled Landslide displacement prediction based on variational mode decomposition and GA-Elman model is a work that presents a landslide displacement prediction method based on a VMD and GA-Elman network, which can achieve good prediction accuracy. This is a meaningful study, but the experimental part can still be improved. I split my comments into two categories below in decreasing order of importance.

Major issues:

-  In section 4.5, the Authors say that compared with Model Two, the combined model performed better, but actually, it has the lowest R2 and a higher RMSE in Table 6, please explain it.

- VMA is a crucial algorithm for the decomposition of cumulative displacement and influencing factors, as mentioned in Line 134, the penalty parameter and the rising step are its two free parameters. So how did the authors set their values and how do they affect the results?

- In section 4.4, the experiments show that the random displacement term can also be predicted with very high accuracy, but to my knowledge, random factors are usually unknown, random or unpredictable. Please explain it.

Minor issues:

- Line 13: “variational model decomposition” should be “variational mode decomposition”?

- Are the titles of Section 2.1 and Section 2.2 the same? “Variational modal decomposition” should be “Variational mode decomposition”?

- Figures 8, 9, and 10 should also draw the measured displacement of the test sets for comparison.

Author Response

Dear Editors and Reviewers,

Thank you very much and the reviewers for your careful and timely review and valuable comments on our manuscript (Manuscript ID: applsci-2066533). Those comments are all valuable and very helpful for revising and improving our paper, as well as the important guiding significance to our researches. We carefully considered and revised the article based on the opinions of reviewers. The revised manuscript of the paper has been revised in review mode and the important revisions are marked in red. The main corrections in the paper and the responds to the reviewer’s comments are as flowing:

Point 1: In section 4.5, the Authors say that compared with Model Two, the combined model performed better, but actually, it has the lowest R2 and a higher RMSE in Table 6, please explain it.

Response 1: We are very sorry for the error in our content. This is a misunderstanding due to a mistake in the writing of the paper. R2 and RMSE values are in the wrong place in Table 6. In fact, R2 is 0.9893 and RMSE is 0.0384.

Point 2: VMD is a crucial algorithm for the decomposition of cumulative displacement and influencing factors, as mentioned in Line 134, the penalty parameter and the rising step are its two free parameters. So how did the authors set their values and how do they affect the results?

Response 2: Thanks to the reviewer for the suggestion. We conducted several comparison experiments during the decomposition. Based on the experimental results the penalty parameter  was set to 1 and the rising step  was set to 0. During the experiment, the penalty parameter and the rising step have a great influence on the decomposition results of the data, especially in the trend term, periodic term and random term of the observed displacement. The results decomposed by different parameters are quite different. After many comparisons, it is finally determined that the observation results are ideal when , , , and .

Point 3: In section 4.4, the experiments show that the random displacement term can also be predicted with very high accuracy, but to my knowledge, random factors are usually unknown, random or unpredictable. Please explain it.

Response 3: Thank you for the suggestion. According to available research results, the random displacement term of landslide is indeed influenced by a variety of unknown factors. Some of the unknown factors that can be foreseen include fluctuations in reservoir levels, precipitation, human activity, earthquakes, etc. In this paper, the grey relational degree method is used to analyse other factors related to the random displacement term. The analysis results show that the random term displacement is closely related to the random term of reservoir water level and precipitation. Therefore, based on the above model optimization and existing data analysis, the prediction accuracy of random displacement is improved. However, the accuracy is not as high as that of the trend and periodic term, indicating that there are other unknown factors influencing the random displacement term of the landslide that need further in-depth study.

Point 4: Line 13: “variational model decomposition” should be “variational mode decomposition”?

Response 4: We are very sorry for the error in our content. This has been amended in the revised version.

Point 5: Are the titles of Section 2.1 and Section 2.2 the same? “Variational modal decomposition” should be “Variational mode decomposition”?

Response 5: We are very grateful for the reviewer's comment and suggestion. We are very sorry for the error in our content. The Section 2.1 is “Variational Mode Decomposition” and the Section 2.2 is “Elman Neural Network”.

Point 6: Figures 8, 9, and 10 should also draw the measured displacement of the test sets for comparison.

Response 6: Thanks to the reviewer for the suggestion. We have drawn the measured displacement of the test sets for comparison in Figures 8, 9, and 10.

We tried our best to improve the manuscript and made some changes in the manuscript. These changes will not influence the content and framework of the paper.

We appreciate for Editors/Reviewers’ warm work earnestly, and hope that the correction will meet with approval.

Once again, thank you very much for your comments and suggestions.

Best regards,

Chenhui Wang

Reviewer 2 Report

This paper proposes a displacement prediction model based on variational model decomposition and genetic algorithm optimization of Elman neural network (VMD-GA-Elman). The reviewer thinks the topic discussed in this paper is very important, which is of great significance for landslide prediction. This reviewer sees that a major revision will be needed before being accepted for possible publication. Here are the main comments for the revision.

1. The engineering geological profile of the landslide should be given in 3.1.

2. It is suggested that the monitoring displacement, rainfall and reservoir water level of Baijiabao landslide should be given in the form of attached table.

3. In the VMD model, the mode number K was set as 4. However, in previous studies, landslide displacement is usually divided into three categories according to time series: trend displacement, periodic displacement and random displacement. Every kind of displacement has practical physical meaning. However, the physical meaning of the four kinds of displacements is not clear. What is the basis?

4. In this paper, several impact factors were chosen. The reasons for the selection of these factors need to be supplemented.

5. It is suggested to supplement the parameter settings of the machine learning model.

6. The results show that the prediction accuracy of the displacement prediction model used in this paper is very high. Is there any over fitting phenomenon? How does the author consider this problem?

7. Landslide displacement prediction is closely related to the mechanism of landslide, not just a mathematical problem. In this paper, there is a lack of in-depth analysis of landslide mechanism. How does the author combine the mechanism with the data in the prediction model?

8. There are many articles on landslide displacement prediction, most of which use known displacement to predict known displacement, which is actually a verification. This is a false prediction. What's the significance?

9. The article is not concise enough, especially the methodology part and the conclusion part.

10. The introduction section lacks some important references, such as: 10.1007/s11069-021-04655-3, 10.1007/s10346-017-0883-y, 10.1016/j.earscirev.2019.03.019, 10.1016/j.enggeo.2022.106779

Author Response

Dear Editors and Reviewers,

Thank you very much and the reviewers for your careful and timely review and valuable comments on our manuscript (Manuscript ID: applsci-2066533). Those comments are all valuable and very helpful for revising and improving our paper, as well as the important guiding significance to our researches. We carefully considered and revised the article based on the opinions of reviewers. The revised manuscript of the paper has been revised in review mode and the important revisions are marked in red. In addition, the manuscript has been revised in English by MDPI. The main corrections in the paper and the responds to the reviewer’s comments are as flowing:

Point 1: The engineering geological profile of the landslide should be given in 3.1.

Response 1: Thanks to the reviewer for the suggestion. The engineering geological profile of the landslide has been added to the revised manuscript in 3.1.

Point 2: It is suggested that the monitoring displacement, rainfall and reservoir water level of Baijiabao landslide should be given in the form of attached table.

Response 2: Thanks to the reviewer for the suggestion. We did not find the original monitoring data of Baijiabao landslide in the public information. We listed the relevant literature about Baijiabao landslide in the references. The above two papers help the authors understand the important influence of precipitation and reservoir water level on landslide displacement in the Three Gorges reservoir area.

  1. Yao W, Li C, Zuo Q et al. Spatiotemporal deformation characteristics and triggering factors of Baijiabao landslide in Three Gorges Reservoir region, China. Geomorphology 2019, 343, 34-47.
  2. Xu S, Niu R. Displacement prediction of Baijiabao landslide based on empirical mode decomposition and long short-term memory neural network in Three Gorges area, China. Computers & Geosciences 2018, 111, 87-96.

Point 3: In the VMD model, the mode number K was set as 4. However, in previous studies, landslide displacement is usually divided into three categories according to time series: trend displacement, periodic displacement and random displacement. Every kind of displacement has practical physical meaning. However, the physical meaning of the four kinds of displacements is not clear. What is the basis?

Response 3: Thank you for the suggestion. According to existing studies, landslide displacement is usually divided into three categories according to time series: trend displacement, periodic displacement and random displacement. We also set the modal number K to 3 in the course of the experiment. While most previous studies have used a monthly cycle for data analysis, we have used a weekly cycle for this study. It was tested that when K = 3, the decomposed periodic term was not obvious. Therefore, K=4 is considered for decomposition and the results of the decomposition are processed analytically. However, the ultimate aim is still to decompose the landslide displacement into three sub-series.

Point 4: In this paper, several impact factors were chosen. The reasons for the selection of these factors need to be supplemented.

Response 4: Thanks to the reviewer for the suggestion. Specific modifications are as follows: The trend displacement of the landslide is related to the internal factors of the landslide. The rock and soil composition of the landslide, changes in the internal stress, and changes in geometric shape, along with other factors, affect the trend displacement of the landslide; moreover, these factors often change over time. Therefore, the main factor affecting the trend displacement of the landslide is the monitoring time. It can be seen from Figure 5 that the cumulative displacement increases to varying degrees with the amount of precipitation. A large increase in cumulative displacement occurs when precipitation is abundant, and displacement slows down when precipitation is relatively low, which shows that the cumulative displacement increases more obviously in the rainy season. In addition, according to related research [47], the reservoir water level also has obvious periodicity. When the water level in the reservoir area tends to fall, the water pressure applied to the surface of the slope decreases and the support capacity of the slope is weakened, accelerating the deformation of the landslide; at these times, the accumulated displacement of the landslide appears to rise rapidly. When the water level in the reservoir area rises, the water pressure acting on the surface of the slope is greater than the force of the slope sliding outwards, inhibiting landslide deformation and slowing down the increase in cumulative displacement. The above analysis demonstrates that precipitation and reservoir levels are important external factors that influence landslide deformation.

Point 5: It is suggested to supplement the parameter settings of the machine learning model.

Response 5: Thanks to the reviewer for the suggestion. We have given the specific parameters of the VMD in section 4.1.1 and the GA-Elman model in section 4.2. Through multiple trial calculations, the relevant parameters of GA were set as follows: the number of population iterations was 30, the population size was 30, the crossover probability was 0.3, and the mutation probability was 0.1. In the setting of relevant parameters for the Elman algorithm, the learning rate was 0.5, the momentum factor was 0.9, and the model accuracy was 0.00001.

Point 6: The results show that the prediction accuracy of the displacement prediction model used in this paper is very high. Is there any over fitting phenomenon? How does the author consider this problem?

Response 6: The overfitting phenomenon did occur during our tests, and after research and analysis, we found that the overfitting phenomenon was caused by inaccurate classification of the monitoring dataset, without in-depth consideration of the important influencing factors affecting landslide displacement. Therefore, the overfitting phenomenon occurred when the data set was not adequately prepared in the first stage. In addition, overfitting is very common in machine learning algorithms, and different techniques should be used to obtain the expected fitting effect, which in turn provides a technical reference for solving practical problems.

Point 7: Landslide displacement prediction is closely related to the mechanism of landslide, not just a mathematical problem. In this paper, there is a lack of in-depth analysis of landslide mechanism. How does the author combine the mechanism with the data in the prediction model?

Response 7: Thanks to the reviewer for the suggestion. The revised manuscript adds an in-depth analysis of the landslide mechanism section of Shuizhuyuan landslide. The supplementary content is as follows: The sliding mass is composed of loose Quaternary deposits. The engineering geological profile of the landslide is shown in Figure 3. The bedrock outcrop section in the middle-lower part of the landslide comprises siltstone and silty mudstone of the Middle Triassic Badong Formation (T2b). The exposed stratum in the middle and trailing edge of the landslide is Upper Triassic Xujiahe Formation (T3xj) sandstone. The sliding zone material of the landslide is mainly silty clay, due to the sliding zone being filled with water and softer plastic, which are prone to sliding deformation. Based on the above analysis, the landslide can be characterized as a soil landslide with precipitation—reservoir as the main inducing factor.

Point 8: There are many articles on landslide displacement prediction, most of which use known displacement to predict known displacement, which is actually a verification. This is a false prediction. What's the significance?

Response 8: As stated by the reviewer, there are indeed many articles on landslide prediction and most of the models use known displacement monitoring data to construct landslide displacement prediction models. The significance of landslide displacement prediction is that unknown monitoring data can be predicted from existing monitoring data. In order to verify the reliability and stability of the prediction model, researchers often select known landslide displacement monitoring data to verify the validity of the designed model. In machine learning methods, the training model can be optimized by dividing the training set, the test set and the validation set, with the aim of improving the generalization ability of the model, which can effectively learn autonomously from similar slippery slopes to complete the prediction function. The prediction model is trained continuously using known data from the training set to obtain a more reliable prediction model. The prediction model is unknown to the data in the test and validation sets, and it is through the test and validation sets that the generalization ability of the prediction model is demonstrated.

In practical landslide displacement work, such as creep, step, mutation and other landslide types, it is impossible to predict in advance which model will produce the best prediction effect. Therefore, it is necessary to understand the disaster mechanism of landslides in advance and to study and test a set of methods, configurations and frameworks for the problem in order to select the optimal model to solve practical problem. Only the design of appropriate landslide prediction model can better achieve the landslide displacement prediction analysis and through the displacement model prediction can intuitively understand the general trend of landslide displacement and deformation. Technicians can make landslide deformation prediction, accurately send out early warning and forecast signals, and timely organize personnel evacuation to avoid casualties.

Point 9: The article is not concise enough, especially the methodology part and the conclusion part.

Response 9: Thanks to the reviewer for the suggestion. We have edited the article again, especially the methodology part and the conclusion part.

Point 10: The introduction section lacks some important references, such as: 10.1007/s11069-021-04655-3, 10.1007/s10346-017-0883-y, 10.1016/j.earscirev.2019.03.019, 10.1016/j.enggeo.2022.106779

Response 10: Thanks to the reviewer for the suggestion. It is really true as Reviewer suggested that the introduction section lacks some important references. The papers you mentioned have been added to the References.

  1. Zhang, Y.; Chen, X.; Liao, R.; Wan, J.; He, Z.; Zhao, Z.; Zhang, Y.; Su, Z. Research on displacement prediction of step-type landslide under the influence of various environmental factors based on intelligent WCA-ELM in the Three Gorges Reservoir area. Hazards 2021, 107, 1709–1729.
  2. Miao, F.; Wu, Y.; Xie, Y.; Li, Y. Prediction of landslide displacement with step-like behavior based on multialgorithm optimization and a support vector regression model. Landslides 2017, 15, 475–488. https://doi.org/10.1007/s10346-017-0883-y.
  3. Intrieri, E.; Carlà, T.; Gigli, G. Forecasting the time of failure of landslides at slope-scale: A literature review. Earth-Sci. Rev. 2019, 193, 333–349. https://doi.org/10.1016/j.earscirev.2019.03.019.
  4. Miao, F.; Zhao, F.; Wu, Y.; Li, L.; Xue, Y.; Meng, J. A novel seepage device and ring-shear test on slip zone soils of landslide in the Three Gorges Reservoir area. Geol. 2022, 307, 106779. https://doi.org/10.1016/j.enggeo.2022.106779.

We tried our best to improve the manuscript and made some changes in the manuscript. These changes will not influence the content and framework of the paper.

We appreciate for Editors/Reviewers’ warm work earnestly, and hope that the correction will meet with approval.

Once again, thank you very much for your comments and suggestions.

Best regards,

Chenhui Wang

Round 2

Reviewer 1 Report

The manuscript has been improved after revision, but there are still some issues to be discussed.

-          I still recommend that the Authors should add experiments to show how the penalty parameter and rising step in VMD affect the results.

-          From Figure 8-13, it seems that there is overfitting, and the Authors should explain it in the manuscript.

Author Response

Dear Editors and Reviewers,

Thank you very much and the reviewers for your careful and timely review and valuable comments on our manuscript (Manuscript ID: applsci-2066533). We carefully considered and revised the article based on the opinions of reviewers. The revised manuscript of the paper has been revised in review mode and the important revisions are marked in red. Key corrections in the paper and detailed responses to reviewers ' comments are included in the accompanying documents. 

We tried our best to improve the manuscript and made some changes in the manuscript. These changes will not influence the content and framework of the paper.

We appreciate for Editors/Reviewers’ warm work earnestly, and hope that the correction will meet with approval.

Once again, thank you very much for your comments and suggestions.

Best regards,

Chenhui Wang

Reviewer 2 Report

This paper has been revised as request, which can be published now.

Author Response

Dear Editors and Reviewers,

Thank you very much and the reviewers for your careful and timely review and valuable comments on our manuscript (Manuscript ID: applsci-2066533). Thanks for the reviewer's suggestions. All your suggestions are very important. They have important guiding significance to my thesis writing and scientific research work.

We appreciate for Editors/Reviewers’ warm work earnestly.

Thank you again for your advice, hoping to learn more from you.

Best regards,

Chenhui Wang
